# Neomycin, but Not Neamine, Blocks Angiogenic Factor Induced Nitric Oxide Release through Inhibition of Akt Phosphorylation

**DOI:** 10.3390/ijms232315277

**Published:** 2022-12-03

**Authors:** Raphaël Trouillon, Dong-Ku Kang, Soo-Ik Chang, Danny O’Hare

**Affiliations:** 1Department of Bioengineering, Imperial College London, London SW7 2BP, UK; 2Department of Chemistry, Imperial College London, 80 Wood Ln, London W12 7TA, UK; 3Department of Biochemistry, Chungbuk National University, Cheongju 28644, Republic of Korea

**Keywords:** cells-on-a-chip, drug screening, angiogenesis, nitric oxide, bio-electrochemical methods

## Abstract

Angiogenesis, the growth of new blood vessels, is a critical factor of carcinogenesis. Neomycin and neamine, two drugs blocking the nuclear translocation of angiogenin (ANG), have been proven to inhibit tumour growth in vivo. However, the high toxicity of neomycin prevents its therapeutic use, thus indicating that the less toxic neamine may be a better candidate. Endothelial cells were cultured on a biocompatible multiple microelectrode array (MMA). The release of NO evoked by ANG or vascular endothelial growth factor (VEGF) was detected electrochemically. The effects of neomycin and neamine on ANG- and VEGF-induced NO releases have been investigated. Neomycin totally blocks NO release for concentrations down to the pM range, probably through the inhibition of the Akt kinase phosphorylation, as revealed by confocal microscopy. On the other hand, both ANG- and VEGF-induced NO releases were not significantly hindered by the presence of high concentrations of neamine. The inhibition of the Akt pathway and NO release are expected to lead to a severe decrease in tissue growth and repair, thus indicating a possible cause for the toxicity of neomycin. Furthermore, the data presented here show that ANG- and VEGF-induced NO releases are not dependent on the nuclear translocation of angiogenin, as these events were not abolished by the presence of neamine.

## 1. Introduction

During carcinogenesis, the sustained tissue growth requires a higher rate of nutrient delivery to the tumour. This high metabolic demand spurs angiogenesis, i.e., the sprouting development of new blood vessels from the healthy vasculature to the tumour cells [1]. Malignant tumours are indeed known to release high levels of angiogenic factors to sustain their growth [2]. Hypoxic conditions are a clinical marker of tumour malignity [3,4] as they indicate a faster development, and can also hinder radiotherapy, as the cytotoxicity of this method is based on the formation of radicals from dissolved oxygen. Tumours are nevertheless critically dependent on vasculature for their development. A higher density of blood vessels also increases the likelihood of intravasation and therefore of metastasis [5,6]. The importance of angiogenesis in tumour growth motivates the use of inhibitors of angiogenesis as anti-tumour drugs. Several of these inhibitors are currently being clinically tested [7,8]. However, some of these inhibitors are too toxic for any therapeutic use. For instance, neomycin, an aminoglycoside antibiotic, can inhibit angiogenesis in vivo [9] but also shows a high level of nephrotoxicity [10] and ototoxicity [11]. Neamine, a by-product of the methanolysis of neomycin, is still anti-angiogenic [12,13] and is much less toxic. The anti-angiogenicity of neomycin and neamine is believed to be due to their ability to block the nuclear translocation of angiogenin (ANG). The interactions of this factor with the cell nucleus are indeed critical for angiogenesis, even when this is induced by other molecules [14]. Screening and comparing the effects on the vascular physiology of these two compounds would therefore provide valuable information for drug development.

Nitric oxide (NO) is a ubiquitous biological messenger involved in neuronal communication [15], immune response [16] and vascular physiology [17]. In this last case, NO is synthesized from L-arginine by a specific type of nitric oxide synthase (NOS), the endothelial nitric oxide synthase (eNOS) [18]. NO synthesis and release can be inhibited by the L-arginine analogue NG -nitro-L-arginine methyl ester [19] (L-NAME). A major role of NO in vasculature is to mediate vasodilation [20]. The endothelial cells lining the lumen of the artery can, upon mechanical or chemical stimulation, release NO which will then diffuse to the underlying smooth muscle cells, inducing a decrease in the tonus of the artery wall [21,22]. In the case of chemical stimulation, NO release can be triggered by exposure to some angiogenic factors, such as vascular endothelial growth factor (VEGF) or ANG [23,24,25]. During angiogenesis, NO increases the permeability of the artery wall, thus enabling migration of endothelial cells and the sprouting of a new artery [26]. It has also been shown to have anti-apoptotic properties, thus promoting tissue proliferation [27]. After exposure to a growth factor, NO is usually released via an intracellular cascade, typically involving the PI-3 kinase/Akt mediation pathway [24,25,28]. 

However, and despite its biological relevance, NO can be challenging to measure accurately in biological conditions, largely owing to its short lifetime (1 to 10 s). Fluorescent methods and molecules, such as the diaminofluoresceins, can be used, but are expensive and are prone to chemical interferences [29]. In contrast, NO can be detected electrochemically with simple setups and electrochemical devices [24]. Electrochemistry is also inherently quantitative, thus improving the quality of the measurements, and therefore a strong candidate for accurate NO detection.

In this report, a biocompatible multiple microelectrode array (MMA) has been used to evaluate the effect of neomycin and neamine on VEGF- and ANG-evoked NO release. This device has been successfully used to show the release of NO in endothelial cells after ANG or VEGF stimulation, and to partially disentangle the pathways leading to this release [24,25,30]. This array was coated with fibronectin to promote cell adhesion and prevents fouling of the electrode by proteins and other surface-active species present in the culture medium [31,32]. The results provided by the microfabricated system were corroborated by confocal microscopy. The purpose of this study is to use NO release, from pig aortic endothelial cells (PAEC) cultured on the surface of the fibronectin coated MMA, as a marker of activation of the PI-3 kinase/Akt pathway to examine the influence of neomycin and neamine. The comparison of the results obtained with neamine and neomycin is expected to provide some indications on this phenomenon. Indeed, neomycin inhibits angiogenesis [9], it is known to prevent nuclear translocation, and we have shown that it also inhibits ANG-induced NOS activation, which occurs via the PI-3 kinase/Akt pathway [24]. However, neomycin is also known to inhibit Akt phosphorylation [33] which must be involved in NOS activation. Neamine is not believed to do this. Using the MMA, we can therefore obtain detailed information on the mode of action of these important compounds. The role of Akt in inhibiting apoptosis suggests this may be an important mechanism, in addition to the inhibition of nuclear translocation [34,35]. Furthermore, in the case of ANG, the role of nuclear translocation of ANG in NO release is still unclear and can be elucidated by comparing the results obtained with neamine and neomycin. Beyond the pharmacological data, this work aims at showing that a well-designed electrochemical device can be integrated into a pharmacological protocol. NO is hard to measure in situ, and the proposed MMA opens the way to massively parallelized screening of compounds of pharmacological interest in a functional assay.

## 2. Results

### 2.1. Description of the Electrochemical Assays

The experimental setup is summarized in Figure 1A. The MMA, composed of six gold working electrodes (diameter 30 µm, recessed by 2 µm) and one gold counter electrode, all insulated with silicon nitride, is coated with fibronectin to promote cell adhesion and stabilize the sensor, thanks to the anti-biofouling properties of dry-coated fibronectin [24,30,36]. This setup is completed with an Ag|AgCl (3 M KCl) reference electrode, added for the duration of the measurements only. As shown in Figure 1B, the principle of this method is to add a secretagogue, here VEGF or ANG, to the cell media, thus inducing the release of an electroactive molecule of interest, here NO. NO can then be detected by the underlying electrode. In the case of NO, most of the signal actually arises from nitrite, the by-product of NO reaction with oxygen, and is nevertheless indicative of NO release. NO release is demonstrated via inhibition of NOS with L-NAME.

**VEGF-induced NO release is blocked by neomycin, but not by neamine.** As shown in Figure 1C, after addition of VEGF, for two electrodes showing the same control trace (from the first set of DPV), neomycin inhibited the increase in the 0.9 V peak, but not neamine. This hints that neomycin and neamine have different activities, as neamine does not abolish ANG-induced NO release. The results for this set of experiments are summarized in Figure 2A.

Compared to the control case, where no angiogenic factor is added, the addition of VEGF led to an 18.8% increase in the magnitude of the NO peak (*p* < 0.001). This increase was completely abolished after addition of the NOS inhibitor L-NAME (100 µM, *p* < 0.001). This result clearly indicates that the increase in the 0.9 V peak is due to a higher NOS activity induced by the presence of the angiogenic factor, leading to the release of NO and to a higher oxidation peak. Similarly, in the presence of 20 µM of neomycin, the release of NO is blocked, as indicated by the abolition of the peak increase (*p* < 0.001). On the contrary, in the presence of neamine, no significant difference can be observed from the case where VEGF only is added (*p* = 0.68), thus indicating that neamine does not interfere with NO release. Decreasing concentrations were incubated over the PAEC. Figure 2B shows that concentrations as small as 200 fM of neomycin inhibit NO release (*p* < 0.001).

### 2.2. Neomycin Blocks eNOS Phosphorylation after Exposure to VEGF

The inhibitory effect of neomycin on NO release was confirmed by investigating the phosphorylation of eNOS after exposure of the PAEC to VEGF. Figure 3 shows confocal imaging obtained for different experimental conditions. The molecules of interest were eNOS and p-eNOS, its phosphorylated form on the serine 1177 residue.

These pictures show a high concentration of cytoplasmic p-eNOS after exposure to VEGF. The addition of neomycin, however, completely abolished the phosphorylation of eNOS, as very little fluorescence associated with p-eNOS can be observed, despite a high concentration of eNOS in almost every compartment of the cell. On the other hand, neamine did not prevent this phosphorylation, as high levels of p-eNOS can be detected in the cytoplasm, as observed when only VEGF is added to the media. These results demonstrate that neomycin, but not neamine, inhibits the phosphorylation of eNOS, as indicated by the results obtained with the MMA.

### 2.3. The Activation of Akt Is Inhibited by Neomycin, Not by Neamine

The PI-3 kinase/Akt pathway has been reported to be central in the mediation of eNOS activation during angiogenesis [24]. As shown on the confocal pictures presented in Figure 4, the addition of VEGF, when compared to the control case where no growth factor is added to the cell media, led to an increase in fluorescence associated to the phosphorylation of Akt on the threonine 308 residue (p-Akt) inside and at the vicinity of the nucleus. This phosphorylation was blocked after addition of neomycin but can still be detected in presence of neamine. This indicates that neomycin, not neamine, blocks the phosphorylation of Akt, thus inhibiting the downstream release of NO.

### 2.4. Neamine Does Not Inhibit ANG Evoked NO Release

The effect of neomycin and neamine on ANG-induced NO release was investigated electrochemically. As shown in Figure 5, the addition of ANG induces NO release, as the 0.9 V peak associated to NOS activation increases by 20% after exposure to ANG (*p* < 0.001). Similar to the VEGF experiments, this increase was totally abolished in presence of 100 µM L-NAME (*p* < 0.001), demonstrating that ANG induces NO release in PAEC. Here again, neomycin inhibited the release of NO after ANG stimulation (*p* < 0.001), but not neamine (*p* = 0.72).

## 3. Discussion

Microelectrode array technology was used for the measurement of angiogenic factor-induced NO release in endothelial cells. Using the ratio of the successive peak currents described by Equation (1) facilitates the use of the MMA by defining a value for the level of NO release relative to control. In particular, no calibration is required, thus overcoming several problems inherent to this type of technology, such as inhomogeneity in cell coverage, uncertainty about the cell to electrode distance, diffusional hindrance by cells and cell debris, etc. This method has been found to be very robust for different types of cells [36]. Importantly, this study uses primary cells that were extracted in-house from porcine aortic endothelial cells (pooled from 4–5 aortas). Cell lines can be characterized by a high level of genetic homogeneity. In contrast, with freshly extracted primary cells, a genetic make-up that better recapitulates the conditions of an actual tissue is expected.

The results presented here show that microfabricated electrochemical systems are attractive methods for quick, massively parallel studies of specific compounds on cellular physiology [37,38]. These electrochemical sensors are cheap, amenable to miniaturisation and require much simpler instrumentation than most of the other available biochemical techniques. Furthermore, simplification of the experimental procedure, for instance by avoiding sensor calibrations, can facilitate non-specialist use and dissemination of that technology as a complement to the traditional biological methods.

Neomycin blocks Akt phosphorylation, unlike neamine. As previously reported, VEGF and ANG induce NO release from endothelial cells, and this release is critically dependent on the PI-3 kinase/Akt mediation [23,24,28]; the pathway is known to be critical for the induction of angiogenesis and vascular homeostasis [39]. Our electrochemical assay showed that eNOS activation, and therefore NO release, was inhibited by neomycin but not neamine. This result was confirmed by immunohistochemistry.

As both VEGF- and ANG-induced NO releases are inhibited, in a similar manner, by neomycin, it was assumed that neomycin may interact with the PI-3 kinase/Akt pathway, but not neamine. This was confirmed by confocal imaging, as neomycin completely abolished the phosphorylation of Akt after VEGF stimulation (Figure 3). This result is in agreement with studies published by others, reporting that neomycin decreases Akt phosphorylation in sheared human umbilical vein endothelial cells [33]. In our assay, this inhibition was not observed with neamine, indicating that neamine does not interfere with this angiogenic kinase pathway. As a consequence, this study suggests that neomycin inhibits NO release by blocking the upstream Akt phosphorylation. This difference could account for the higher toxicity of neomycin, compared to neamine.

NO release is independent of the nuclear translocation of angiogenin. Angiogenin is a fundamental factor in angiogenesis. Its actions during blood vessel growth are multiple, as it can:(i)Cleave actin moieties thus inducing cell migration [40];(ii)Activate several kinase pathways [41,42];(iii)Undergo nuclear translocation and presumably interact with ribonucleotides [43];(iv)Show some RNase activity [44,45].

Both neomycin and neamine are known to inhibit angiogenesis, and hence carcinogenesis [9,12,13,46]. These compounds can also inhibit angiogenesis induced by other angiogenic factors, mostly through the inhibition of the nuclear translocation of ANG, a general requirement for blood vessel growth [14]. The interaction of ANG into the nucleus indeed induces the synthesis of small interfering RNAs, mediating neo-vascularization [14].

The release of NO after exposure to ANG is mediated by the activation of the PI-3 kinase/Akt pathway [24], but is independent of the RNase activity of ANG [30]. The exact role of nuclear translocation was still unclear in this phenomenon, mostly because of the toxicity of the neomycin used to inhibit this pathway. As shown in this report, neomycin actually inhibits eNOS activation through inhibition of the Akt pathway, as summarized in Figure 6. On the other hand, neamine is expected to solely interact with the nuclear translocation pathway, thus providing a selective inhibitor of this activity. The data presented in Figure 5 show that neamine has no effect on NO release, and that the nuclear translocation of ANG is independent of the eNOS activation. As shown in Figure 5, after ANG stimulation, the eNOS activation is solely mediated via the PI-3 kinase/Akt pathway, and is independent of the RNase activity and the nuclear translocation of ANG.

The effects of neamine and neomycin have been compared, thus showing that neomycin, but not neamine, blocks NO release via the inhibition of Akt phosphorylation. This result is a potential explanation for the high toxicity of neomycin, as most of the constitutive NO release is virtually blocked, thus probably inhibiting, or at least significantly hindering, tissue growth. Furthermore, the fact that neamine is a selective blocker of the nuclear translocation of ANG was used to show that ANG-induced NO release is independent of the nuclear translocation of ANG and is only mediated through a kinase cascade. More generally, this report supports the use of microfabricated electrochemical devices as a tool for biochemistry [36]. Once the issues of the bio-integration and the bio-stability have been addressed, their low price, simple architecture and ideal format for miniaturization and mass production makes them promising candidates for massively parallel screening of drugs.

## 4. Materials and Methods

Chemicals: deionized water (resistivity > 18 MΩ.cm) from a Millipore system was used for all experiments. All the chemicals were purchased from Sigma, were of analytical grade and used without further purification. The 20 µg.mL^−1^ fibronectin solution was prepared by dissolving fibronectin from bovine plasma in Dulbecco’s modified Eagle’s Medium (DMEM). The 5 µg.mL^−1^ VEGF solution was prepared by dissolving mouse VEGF into water. The bovine ANG was produced as previously described and dissolved in water (1.4 mg.mL^−1^, [47]). Neomycin (neomycin sulphate, Calbiochem) and neamine 20 mM solutions were prepared by dissolving these compounds in water. The basal media used for the electrochemical tests were made of DMEM with 10% (*v*/*v*) new-born calf serum, 5 mM L-glutamine. To obtain the culture media in which the cells were maintained, 5 µg.mL^−1^ endothelial growth factor was added to the basal media (Warboys et al., 2010). All the solutions were stored at 4 °C.

Preparation of neamine: neamine was prepared by methanolysis of neomycin, as described by others [12,48]. Briefly, neomycin sulphate was refluxed in methanol and HCl for 4 h. The resulting neamine was precipitated with anhydrous ether. The mixture was filtered, and the filtrate was washed twice with ether. The final product was a white, crystalline powder of neamine.

Cell culture: PAEC were harvested from freshly excised aortas using collagenase and cultured in culture medium at 37 °C, 95% O_2_, 5% CO_2_, in gelatine coated flasks [49]. The growth media were changed every other day and the cells were passaged every week. Cells from passage 2 to 6 were used.

Modification and preparation of the sensors: the sensors were prepared as previously described [23,30,39]. The MMA were modified using a Sylgard (Dow Corning) custom-made reaction cell to allow deposition of a 500 µL volume on the sensor. A lid made from a Petri dish was also fitted on the cell to minimise evaporation and avoid contamination when the sensor had to be placed in the humid incubator. Prior to any experiment, biological debris was removed using trypsin solution. Trypsin was deposited into the cell and incubated at 37 °C for one hour. They were then rinsed with 70% ethanol followed by water. The gold electrodes were electrochemically cleaned by performing cyclic voltammograms between 1.6 and −0.3 V vs. Ag|AgCl at 0.5 V.s^−1^ in 0.1 M sulfuric acid until reproducible voltammograms were obtained. The sensor was then sterilized with 70% ethanol, placed into the sterile safety hood and rinsed with PBS (pH = 7.4). A 50 µL-drop of bovine fibronectin solution was deposited on the sensor and let to dry. The excess of fibronectin was then rinsed with PBS.

Electrochemical measurements: cells were harvested and counted with a cytometer and Trypan blue. A total of 1000 cells were suspended in 500 µL of culture media and deposited on the sensor. The chip was incubated overnight (37 °C, 5% CO_2_). The following day, the culture media were replaced with 500 µL of basal media. For some experiments, the cells were pre-incubated with neomycin, neamine or L-NAME. These inhibitors were added at least one hour before the first measurements to obtain a final concentration of 20 µM for neomycin and neamine, and 100 µM for L-NAME. However, as a set of experiments, the concentration of neomycin was gradually decreased from 20 µM to 200 fM. A set of three successive differential pulse voltammograms (DPVs, pulse width = 0.06 s, amplitude = 0.05 V, increment = 0.004 V) was recorded between 0.4 V and 1.2 V vs. Ag|AgCl to provide a control trace. ANG or VEGF were then added at relevant final concentrations, 100 ng.mL^−1^ [23] and 5 µg.mL^−1^ [24], respectively. The MMA was placed in the incubator and another set of two DPV was recorded after 2 h (VEGF) or 1 h (ANG).

Data processing: for data processing, the third DPV of the first set (before addition of angiogenic factor) and the second DPV of the second set (after addition of angiogenic factor) were used. This sequence was experimentally found to provide stable and reproducible results. The DPV were analysed by normalizing the peak current seen at 0.9 V vs. Ag|AgCl obtained after VEGF or ANG injection by the peak current measured before injection:(1)$$Current\;ratio=\frac{\left(peak\;current\;after\;ANG\;or\;VEGF\right)}{\left(peak\;current\;before\;addition\right)}$$

This normalization guaranteed consistency between the different measurements by minimizing the effect of variations in size, background species, heterogeneity in cell population, etc. [23]. The results obtained for each experimental condition, from several MMA, were then averaged. The standard deviation was calculated. To accurately report the process variability, the number of data points *n* is presented as the total number of measurements with the number of MMA used between brackets: *n* = number of measurements (number of MMA).

Each experimental situation was compared to the case where only the angiogenic factor (ANG or VEGF) is added, to assess the level of inhibition. Assuming the normality of the results obtained from the MMA, each dataset was compared to the stimulated response using Student’s *t*-tests.

Laser Confocal Microscopic Analysis: cells were plated sparsely (5 × 103 cells.cm^−2^) on 18 × 18 mm glass slides coated with fibronectin. The cultures were serum-deprived overnight with basal serum-free media. If required, a 30 min pre-treatment with inhibitors (50 µM neomycin or 50 µM neamine) was performed, and VEGF was added at a final concentration of 50 ng.mL^−1^. Incubation was continued for 30 min. Cells were fixed in 100% methanol for 5 min at −20 °C and rinsed three times with cold PBS for 5 min at room temperature. The cells were then incubated with 1% bovine serum albumin in PBS at room temperature. The cells were then treated for 1 h at 37 °C with a set of mouse anti-eNOS antibodies and rabbit anti-phospho-eNOS (Ser1177) antibodies (Cell Signalling Technology) or a set of mouse anti-Akt antibodies and rabbit anti-phospho-AKT (Thr308) antibodies (Cell Signalling Technology). After being rinsed with PBS, anti-mouse Cy3-labeled antibodies and anti-rabbit Cy5-labeled antibodies (Sigma-Aldrich) were incubated for 1 h at room temperature and then rinsed five times for 5 min with PBS at room temperature. Slides were mounted using gel mount (Biomedia) on microslides (Paul Marienfeld GmbH & Co KG), and the cells were observed with an inverted confocal imaging system (Leica TCS SP5).

## Figures and Tables

**Figure 1 ijms-23-15277-f001:**
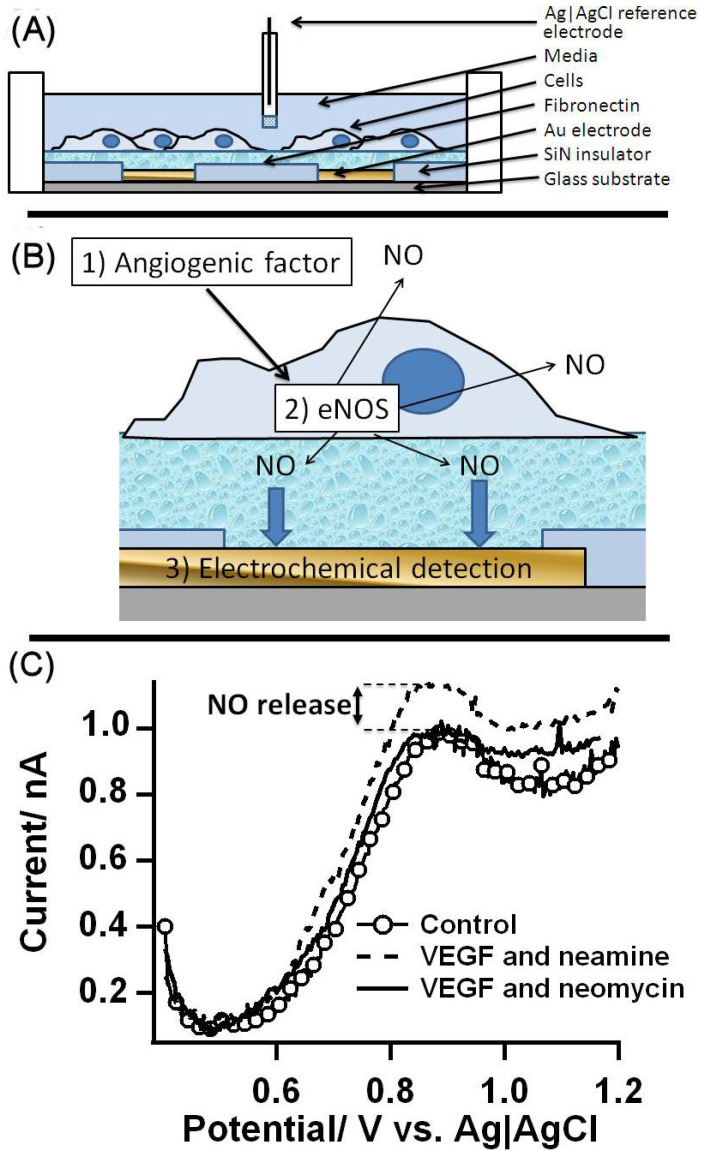
Principle of the electrochemical assay. (**A**) Scheme of the experimental setup, showing the endothelial cells, maintained in their culture media, and grown on a fibronectin sensor array. (**B**) Principle of the measurements, where the angiogenic factor induces NO release, which can, or its oxidation product nitrite, be detected at the underlying electrodes. (**C**) Typical traces obtained for DPV performed in different conditions: control DPV at t = 0 h, t = 2 h after addition of 100 ng.mL^−1^ of VEGF with 20 µM neomycin (—) or 20 µM neamine (- - -).

**Figure 2 ijms-23-15277-f002:**
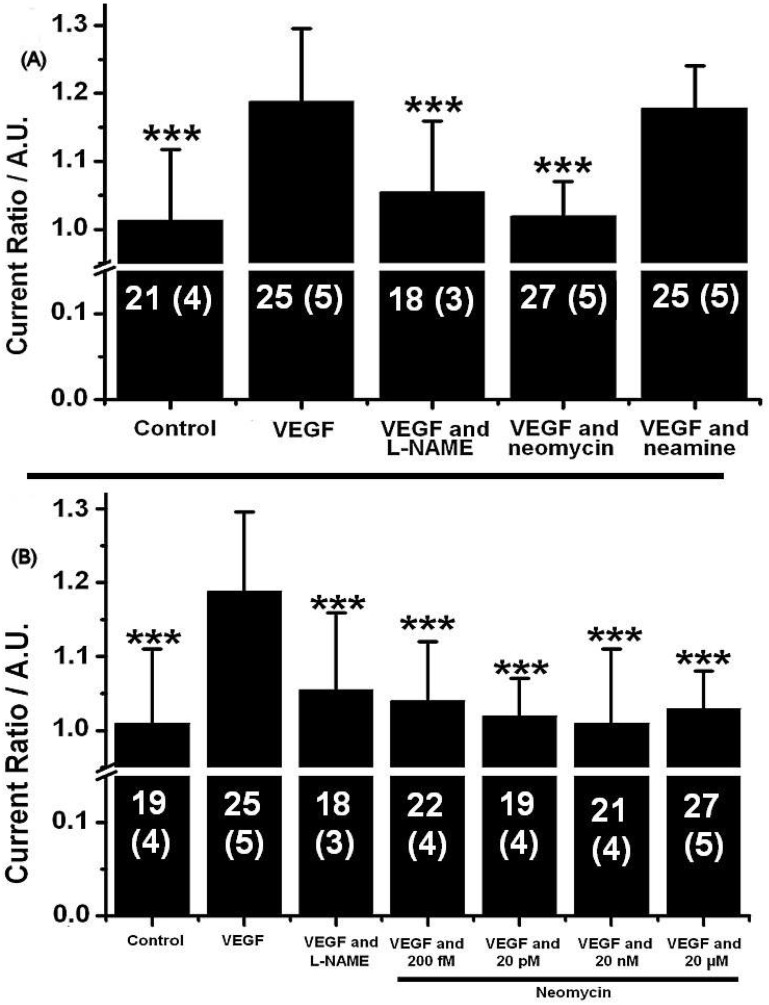
Effects of neomycin and neamine on VEGF-induced NO release. (**A**) Current ratios obtained for a 2 h exposure to 100 ng.mL^−1^ of VEGF in the presence of different inhibitors (neomycin, neamine, L-NAME). (**B**) Current ratios obtained for a 2 h exposure to 100 ng.mL^−1^ of VEGF in the presence of different concentrations of neomycin. The data shown here are average standard deviation. The graph shows the total number of measurements with the number of MMA used between brackets, for each condition: n = number of measurements (number of MMA). The values obtained were compared to the control case, where only VEGF was added using Student’s *t*-test, ***: *p* < 0.001.

**Figure 3 ijms-23-15277-f003:**
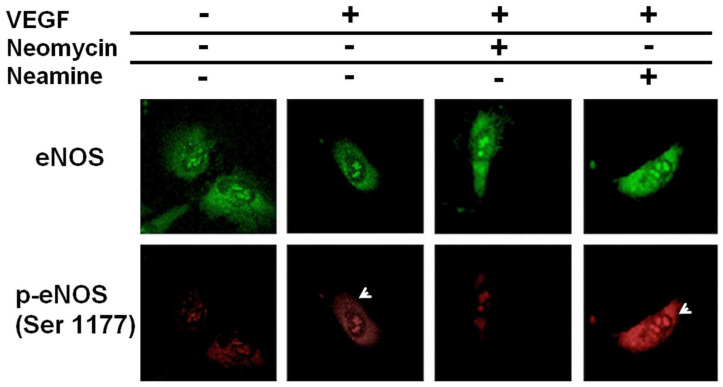
Confocal imaging of eNOS activation. The images obtained for eNOS are shown in green (top row), the ones obtained for its phosphorylated form on the serine 1177 residue, p-eNOS, in red (bottom row). Neomycin and neamine were used as inhibitors. The white arrows indicate the sites where a high level of p-eNOS is observed, in comparison to the control case.

**Figure 4 ijms-23-15277-f004:**
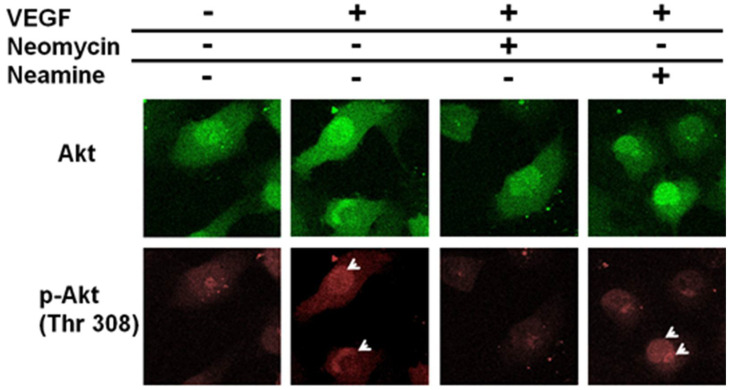
Confocal imaging of Akt activation. The images obtained for Akt are shown in green (top row), the ones obtained for its phosphorylated form on the threonine 308 residue, p-Akt, in red (bottom row). Neomycin and neamine were used as inhibitors. The white arrows indicate the sites where a high level of p-Akt is observed, in comparison to the control case.

**Figure 5 ijms-23-15277-f005:**
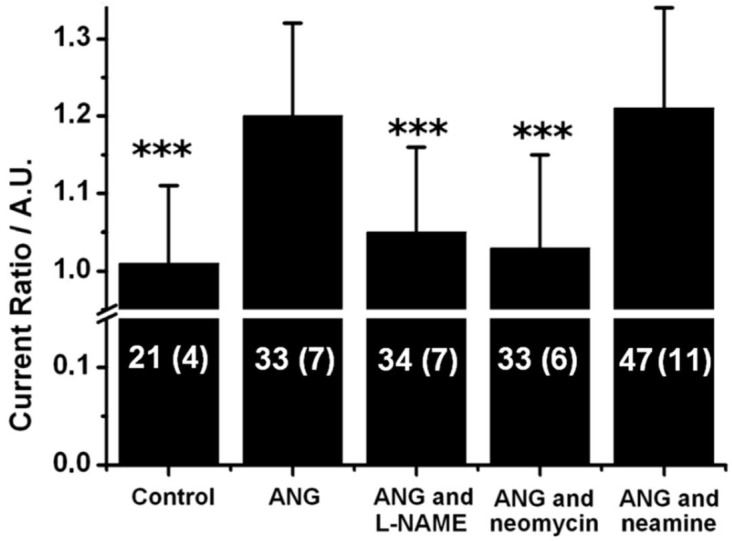
Effects of neomycin and neamine on ANG-induced NO release. Current ratios obtained for a 1 h exposure to 5 µg.mL^−1^ of ANG in presence of different inhibitors (20 µM neomycin, 20 µM neamine, 100 µM L-NAME). The data shown here are average ± standard deviation. The graph shows the total number of measurements with the number of MMA used between brackets, for each condition: n = number of measurements (number of MMA). The values obtained were compared to the control case where only VEGF was added using Student’s *t*-test, ***: *p* < 0.001.

**Figure 6 ijms-23-15277-f006:**
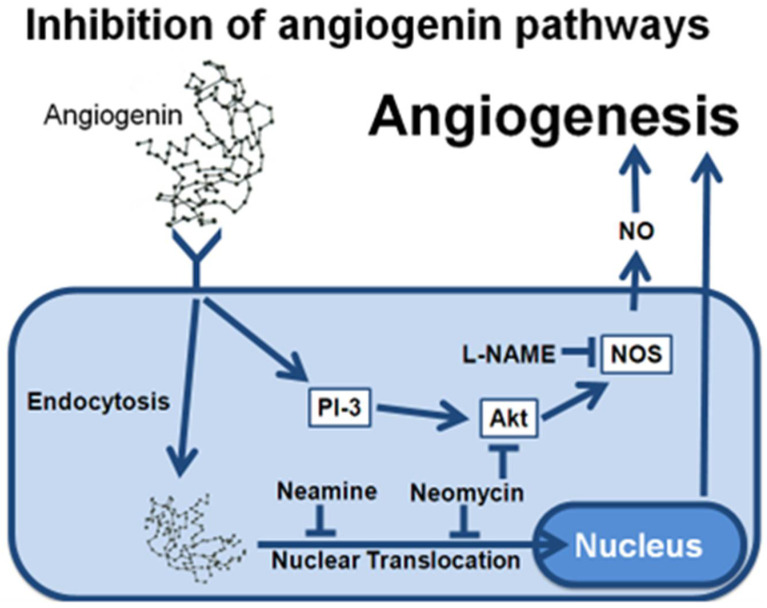
Summary of the role of the different inhibitors in ANG-induced NO release.

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
