# Peer review of "Neomycin, but Not Neamine, Blocks Angiogenic Factor Induced Nitric Oxide Release through Inhibition of Akt Phosphorylation"

_ijms, 2022, doi:10.3390/ijms232315277_

Round 1

Reviewer 1 Report

The article Neomycin, but not neamine, blocks angiogenic factors induced nitric oxide release through inhibition of Akt phosphorylation presents some interesting findings refilling our knowledge about angiogenesis development in cancer progress.

The authors used relatively narrow portfolio of experimental technics, but unfortunately only one experimental cell line was used in experiments. In my point of view, two experimental cell lines should be used at least. There is no doubt that the research in the field of angiogenesis belongs to the most studied issues in cancer research in last forty years. Based on this fact, it is important to set the presented information into the wide conception. I strongly recommend to include some other key results presented in the field of cancer angiogenesis research.

The quality of graphs presented in figures 1 and 4 must be improved. The resolution of graphs and its description must be improved as well. It is not possible to clearly understand the meaning of numbers in particular columns appearing in the graphs. In the same line, the confocal microscopy analyses presented as figures 2 and 3 do not show appropriate visual quality and therefore it is very problematic to observe the mentioned phenomena (The resolution, contrast and size of figures must be altered). The chapter 2.6 Data processing lacks the information on the number of independent experiments realized. Please, fill in the missing information.

Author Response

1) The authors used relatively narrow portfolio of experimental technics, but unfortunately only one experimental cell line was used in experiments. In my point of view, two experimental cell lines should be used at least.

Response & action taken:

The purpose of the paper is to demonstrate the possibility of using a MMA-based approach to obtain precise pharmacological data. From this perspective, the critical dataset of the paper is the electrochemical data, the additional experiments supporting our findings. This is highlighted in the last paragraph of the Introduction

Action taken: The following paragraph was added (page 3)

“Beyond the pharmacological data, this work aims at showing that a well-designed electrochemical device can be integrated into a pharmacological protocol. NO is hard to measure in situ, and the proposed MMA opens the way to massively parallelized screening of compounds of pharmacological interest, in a functional assay.’

We did not use actual cell lines, but primary cells that were extracted in-house from porcine aortic endothelial cells. The purpose is here to overcome the homogeneity of cell lines which is expected to limit the relevance of the assay. Here, by using cells extracted from actual tissue, and pooled from 4-5 aortas, we can guarantee a level of genetic diversity that better recapitulates the conditions of an actual endothelium.

Action taken: The following statement was added (page 8)

“Importantly, this study uses primary cells that were extracted in-house from porcine aortic endothelial cells (pooled from 4-5 aortas). Cell lines can be characterized with a high level of genetic homogeneity. In contrast, with freshly extracted primary cells, a genetic make-up that better recapitulates the conditions of an actual tissue is expected.”

2) The chapter 2.6 Data processing lacks the information on the number of independent experiments realized

Response & Action:

The number of experiments was explained in the Data Processing subsection of the Materials and Methods section (page 10):

“To report accurately the process variability, the number of data points n is presented as the total number of measurements with the number of MMA used between brackets: n = number of measurements (number of MMA).”

Action taken: A statement explaining the reported number of data points is added to the caption of Figures 2 and 5

Higher resolution images are available to address referee comments and can be uploaded on request.

Reviewer 2 Report

The manuscript by Raphael Trouillon, et al. on “Neomycin, but not neamine, blocks angiogenic factor induced nitric oxide release through inhibition of Akt phosphorylation”, compare the effect of neomycin and neamine and confirm the inhibition of neomycin on NO release through the inhibition of the Akt kinase phosphorylation by using different experiments. The manuscript looks good in whole, but there are still some issues to be illustrated before it is fully accepted for publication.

1.      On page 1, the authors mention “the high toxicity of neomycin prevents its therapeutic use”. Then why do the authors pick neomycin for these studies and what’s the prospects of neomycin?

2.      Although “background and purpose” section states “the less toxic neamine may be a better candidate”, the paper focus on the effect of neomycin and confirms the inhibition of neomycin on NO release. I feel necessary additional experiments for the authors to study the effect mechanism of neamine as an angiogenesis inhibitor.

3.      To give readers a good understanding, the authors should clearly state the meaning of 21 (4) and 25 (5) in figure 2.  

4.      In 3.3 section, the authors mention “The PI-3 kinase / Akt pathway has been reported to be central in the mediation of eNOS activation during angiogenesis”, but supply no reference. I feel necessary additional experiments for the authors to study the link between PI-3 kinase / Akt pathway and eNOS activation in pig aortic endothelial cells.

5.      Confocal images only contain one or two cells. The authors should supply images containing more cells.

Author Response

1) On page 1, the authors mention “the high toxicity of neomycin prevents its therapeutic use”. Then why do the authors pick neomycin for these studies and what’s the prospects of neomycin?

Response & Actions

This report is part of a series of paper focused on NO release from AG stimulation. In this field, the nuclear translocation of ANG is typically inhibited with neomycin. However, owing the high toxicity of neomycin, we cannot guarantee that the observed results are due to this specific inhibition. The less toxic neamine is here tested as an alternative.

2) “the less toxic neamine may be a better candidate”, the paper focus on the effect of neomycin and confirms the inhibition of neomycin on NO release. I feel necessary additional experiments for the authors to study the effect mechanism of neamine as an angiogenesis inhibitor.

Response & action taken:

The important aspect of the study is actually the comparison of the neomycin and neamine data. This is clearly shown in Fig1A (before, Fig6A). Both neomycin and neamine are known to inhibit ANG nuclear translocation. Their impact on NO release are different, as neamine does not block NO secretion in response to ANG stimulation.

The important result is here that neamine is a more specific inhibitor of nuclear translocation than neomycin.

Action taken: To better highlight the neomycin/ neamine comparison, Fig 6 was brought to the beginning of the paper as Figure 1. A subsection (2.1) explaining the experimental setup was added, with the following statement (page 5)

“This hints that neomycin and neamine have different activities, as neamine does not abolish ANG-induced NO release”

2) the authors should clearly state the meaning of 21 (4) and 25 (5) in figure 2

This is actually the number of experiments, as explained in the Data Processing subsection of the Materials and Methods section (page 10):

“To report accurately the process variability, the number of data points n is presented as the total number of measurements with the number of MMA used between brackets: n = number of measurements (number of MMA).”

Action taken: A statement explaining the reported number of datapoints is added to the caption of Figures 2 and 5.

3) the authors mention “The PI-3 kinase / Akt pathway has been reported to be central in the mediation of eNOS activation during angiogenesis”, but supply no reference. I feel necessary additional experiments for the authors to study the link between PI-3 kinase / Akt pathway and eNOS activation in pig aortic endothelial cells.

This was actually reported in Ref 24, as mentioned in the Introduction

Action taken: A reference to paper 24 was added page 6.

3) Confocal images only contain one or two cells. The authors should supply images containing more cells.

The confocal data is here used as a confirmatory experiment, to support the differential inhibition of the PI-3/ Akt pathway after exposure to neamine or neomycin. Images with more cells would inevitably require lower magnification and would not adequately show the localisation of the labelled proteins. Finally, angiogenic responses requires lower cell density than confluence in primary endothelial cells.

Reviewer 3 Report

Neomycin, but not neomine, blocks angiogenic factor induced nitric oxide release through inhibition of Akt Phosphorylation 

The authors investigate two molecules neomycin and neomine for their anti-angiogenic properties and mechanisms using a microelectrode array. Authors through their studies show that neomycin blocks the NO production vital for angiogenesis while neomine does not show any hindrance even at higher concentrations. Authors have done a detailed investigation on how the molecules interact with the pathways leading to NO production. The manuscript is well written, and use of new technologies can improve future drug screening for therapeutic purposes in a high throughput manner. Although, few concerns need to be addresses before acceptance for publication. Please find below listed concerns 

  1. 1. Abstract need to mention the numbers (fold change, etc) by which neomycin inhibits NO and what higher concentrations of neomine were used. Please also add a statement on the use of MMA employed in this study 

  1. 2. Introduction needs to compare similar technologies related or equivalent to MMA to show the significance of use of this technology. A short paragraph introduction should suffice. 

  1. 3. Explanation of choice of pig endothelial cells against use of human endothelial cells is missing, which needs to be addressed. 

  1. 4. In results section 2.1 please correct Figure 1c (line 94) as there is no figure 1c to be found in the manuscript 

  1. 5. Rationale for 2-hr time period vs longer times of exposure of drugs may be explained here. 

  1. 6. Figure 2 needs to be improved with quantitative data from multiple cells and plotting mean fluorescence.  

  1. 7. Additional study supporting (Western blot/other PCR techniques) Figure 3 confocal images  with similar conditions (shear) or study with human endothelial cells is needed to compare and draw parallels with studies pointed out in the literature (line 199). 

  1. 8. Figure 6: Images of actual MMA and/or cell seeded array can show the footprint and glimpse of the technology for better understanding of the reader. Highly recommend adding real pictures of the setup. 

Author Response

1) Abstract need to mention the numbers (fold change, etc) by which neomycin inhibits NO and what higher concentrations of neomine were used. Please also add a statement on the use of MMA employed in this study 

Action taken: The requested details have been added to the abstract.

2) Introduction needs to compare similar technologies related or equivalent to MMA to show the significance of use of this technology. A short paragraph introduction should suffice. 

Action taken: The following paragraph was added in the Introduction (page 2)

“However, and despite its biological relevance, NO can be challenging to measure ac-curately in biological conditions, largely owing to its short life-time (1 to 10s). Fluores-cent methods and molecules, such as the diaminofluoresceins, can be used, but are ex-pensive and are prone to chemical interferences [ ]. In contrast, NO can be detected electrochemically with simple setups and electrochemical devices [24]. Electrochemistry is also inherently quantitative, thus improving the quality of the measurements, and therefore a strong candidate for accurate NO detection.”

3) Explanation of choice of pig endothelial cells against use of human endothelial cells is missing, which needs to be addressed.

Response: The reason is technical, as we had access during the study to a steady supply of pig aortas. It is also a cost-effective solution.

4) In results section 2.1 please correct Figure 1c (line 94) as there is no figure 1c to be found in the manuscript.

Response: Indeed, there was a mistake in the ordering of the Figures, thank you very much for pointing this out. The former Fig 6 is now Fig 1, as it better explains the MMA protocol.

5) Rationale for 2-hr time period vs longer times of exposure of drugs may be explained here

The protocol is actually derived from Ref 24.

NO release is typically a short event, rapidly mediated via kinase phosphorylation. To better accommodate this time-scale, we chose a 2-hr incubation time.

6) Figure 2 needs to be improved with quantitative data from multiple cells and plotting mean fluorescence

The confocal data is here used as qualitative confirmatory experiments, to support the differential inhibition of the PI-3/ Akt pathway after exposure to neamine or neomycin.

7) Additional study supporting (Western blot/other PCR techniques) Figure 3 confocal images  with similar conditions (shear) or study with human endothelial cells is needed to compare and draw parallels with studies pointed out in the literature (line 199).

We have published a series of paper focused at ANG- or VEGF-induced NO release (refs 23, 24, 30, 36). Some of these reports were based on HUVECs, and show scratch migration assays. The proposed ANG pathway is derived from these studies and from several techniques, on different cell lines.

8) Figure 6: Images of actual MMA and/or cell seeded array can show the footprint and glimpse of the technology for better understanding of the reader. Highly recommend adding real pictures of the setup.

We have already published such images in the cited papers, references 23 and 24.

Round 2

Reviewer 1 Report

No comments to be added.